# Alterations of Phenylpropanoid Biosynthesis Lead to the Natural Formation of Pinkish-Skinned and White-Fleshed Strawberry (*Fragaria × ananassa*)

**DOI:** 10.3390/ijms23137375

**Published:** 2022-07-01

**Authors:** Leiyu Jiang, Maolan Yue, Yongqiang Liu, Yuyun Ye, Yunting Zhang, Yuanxiu Lin, Xiaorong Wang, Qing Chen, Haoru Tang

**Affiliations:** 1College of Horticulture, Sichuan Agricultural University, Chengdu 611130, China; jiangleiyusicau@outlook.com (L.J.); maolanyue@outlook.com (M.Y.); liuyq0129@163.com (Y.L.); yyycd92@163.com (Y.Y.); wangxr@sicau.edu.cn (X.W.); 2Institute of Pomology & Olericulture, Sichuan Agricultural University, Chengdu 611130, China; asyunting@sicau.edu.cn (Y.Z.); linyx@sicau.edu.cn (Y.L.)

**Keywords:** anthocyanin, strawberry flesh, transcriptomic and metabolomics, *FaC4H*, promoter methylation

## Abstract

Anthocyanin content is important for both the external and internal fruit quality of cultivated strawberries, but the mechanism of its accumulation in pinkish-skinned and white-fleshed strawberries is puzzling. Here, we found that the factor determining variation in the flesh color was not the *FaMYB10* but the *FaC4H* in the cultivated strawberry Benihoppe and its white-fleshed mutant Xiaobai. Compared with Benihoppe, there was no significant difference in the coding sequence and expression level of *FaMYB10* in Xiaobai’s flesh. Instead, the transcription of *FaC4H* was dramatically inhibited. The combined analyses of transcriptomics and metabolomics showed that the differential genes and metabolites were significantly enriched in the phenylpropanoid biosynthesis pathway. Furthermore, the transient overexpression of *FaC4H* greatly restored anthocyanins’ accumulation in Xiaobai’s flesh and did not produce additional pigment species, as in Benihoppe. The transcriptional repression of *FaC4H* was not directly caused by promoter methylations, lncRNAs, or microRNAs. In addition, the unexpressed *FaF3′H*, which resulted in the loss of cyanidin 3-O-glucoside in the flesh, was not due to methylation in promoters. Our findings suggested that the repression of *FaC4H* was responsible for the natural formation of pinkish-skinned and white-fleshed strawberries.

## 1. Introduction

Cultivated strawberry (*Fragaria × ananassa*, 2n = 8x = 56) is grown all over the world, and its fruits are rich in nutrients and flavonoids; hence, they are deeply loved by consumers. As one of the flavonoid metabolites, anthocyanin is crucial to the quality of strawberry fruits. Usually, there are three different cultivated strawberry varieties: red-skinned and red-fleshed, white-skinned and white-fleshed, and pinkish-skinned and white-fleshed. However, the factors that cause changes in anthocyanins are puzzling, especially for the latter variety.

The complete biosynthesis process of flavonoids has been well-studied in strawberries, including the general phenylpropanoid pathway, and a specific flavonoid branch [1,2,3]. In the general phenylpropanoid pathway, phenylalanine acts as a precursor, undergoing a series of enzymatic reactions to finally generate p-coumaroyl-CoA. The genes involved in this process include *PAL*, *C4H*, and *4CL* [4]. In addition, the phenylpropanoid pathway is also closely related to hydroxycinnamic acids, osthole, eugenol, cinnamaldehyde, lignin, dihydrochalcones and ubiquinone biosynthesis [5,6,7]. In the specific flavonoid pathway, naringenin chalcone produces flavones, isoflavones, flavonols, anthocyanins, and flavan-3-ols (precursors of proanthocyanidin polymers) using the enzymatic reaction of different branches [8]. The main anthocyanins are pelargonidin (>80%) and cyanidin in the red fruit of cultivated strawberry subspecies, which give the fruit its bright-red and dark-red appearance, respectively [9,10].

Similar to other plants, the regulation of strawberry anthocyanins and proanthocyanidins (PAs) is also modulated by the ternary complex MYB-bHLH-WD40 (MBW), in which the R2R3-MYB transcription factor (TF) plays a pivotal role [2,11]. The FaMYB9/FaMYB11-FabHLH3-FaTTG1 complex positively regulates the PAs metabolism. The MBW, with FaMYB10 as the core, may positively regulate the anthocyanins metabolism in strawberry fruit, while FaMYB1 represses both processes [2,11,12,13,14,15]. Moreover, the FaRAV1 TF contributes to anthocyanin biosynthesis by upregulating anthocyanin-related genes, including *FaMYB10* [16]. Some reports also demonstrated that small RNAs such as miR829.1, miR1873, miRNA858a, and miRNA156 can regulate flavonoid metabolism by targeting the structural genes [17,18].

Generally, mutations in the coding region or promoter sequence of the structural genes or flavonoid-related TFs are key to impairing pigment accumulation in horticultural fruits [19,20,21]. Until now, the allelic variations in *MYB10* are believed to be the main driving force for the differential distribution of anthocyanins in wild and cultivated strawberry fruits under natural conditions [22,23]. An identified SNP in the *FveMYB10* coding sequence leads to the formation of yellow fruits of wild strawberries through genome-scale DNA variant analysis [24]. The white-specific variant *FaMYB10-2*, an ACTTATAC insertion in *FaMYB10*, is responsible for the pigment deficiency in the skin and flesh of cultivated strawberries [14,23]. Moreover, a *FaEnSpm-2* (CACTA-like) transposon is always located on the *MYB10-2* promoter in the red-fleshed strawberries, which increases the expression of *MYB10-2* and anthocyanin-related genes [23]. Therefore, a large amount evidence for the natural variation in strawberry fruit color converges on the loss of function or differential expression of the *MYB10* gene.

Xiaobai is a certified strawberry variety derived from the somatic variation of Benihoppe virus-free seedlings under tissue culture conditions. It is very popular due to its better flavor and economic benefits [25]. However, it is not clear how the red-fleshed Benihoppe mutates into the white-fleshed Xiaobai. Since Benihoppe, Xiaobai, and most of the octoploid strawberries are vegetatively propagated, it is very difficult to locate the mutation site by QTL mapping. The incompleteness of the octoploid strawberry genome data also makes it impossible to use whole-genome sequencing methods to find mutated genes, as for the diploid strawberry [26]. In this study, we found that the change in flesh color in octoploid strawberry cultivars Benihoppe (red-skinned and red-fleshed) and Xiaobai (pinkish-skinned and white-fleshed) was not ascribed to the dysfunction of *FaMYB10*, but to the differential expression of *FaC4H*. We performed transcriptomic, metabolomics, RT-qPCR, and HPLC assays for the flesh of Benihoppe, Xiaobai, 35S::FaMYB10 (restore anthocyanin accumulation in Xiaobai), and 35SN (control). In addition, *FaC4H* was also transiently overexpressed in Xiaobai fruit. Furthermore, the bisulfite sequencing was used to explore the methylation levels of the *FaC4H* and *FaF3′H* promoters in the skin and flesh of Benihoppe and Xiaobai. These results provided new details to further the understanding of the mechanism of pigment accumulation in the pinkish-skinned and white-fleshed strawberries.

## 2. Results

### 2.1. Variations in Anthocyanin-Related Genes in the Flesh of Benihoppe and Xiaobai Strawberries

As shown in Figure 1a, the flesh of Benihoppe accumulated pigments, while its bud mutant cultivar Xiaobai (variety authorization number: CNA20141360.2) did not [25]. Further HPLC analysis uncovered that the main anthocyanins in Benihoppe’s flesh were pelargonidin 3-O-glucoside (Pg3G) with 306 μg/g fresh weight (FW). This was not detected in Xiaobai (Figure 1b). Moreover, the anthocyanins content in Benihoppe’s skin was also about four times higher than that in Xiaobai, in which Pg3G and cyanidin 3-O-glucoside (Cy3G) were 487 μg/g FW and 44 μg/g FW, respectively (Figure 1b). Benihoppe’s fruits were accompanied by a higher PAs content (Appendix A). The qPCR results showed that the expression levels of most structural genes in the whole flavonoid biosynthesis pathway were higher in the Benihoppe’s flesh than those in Xiaobai, especially for *C4H*, *F3H*, *ANR*, *F3GT*, and *TT19* (Figure 1c–e). Among these genes, *C4H* and *TT19* changed by about 27-fold and 17-fold, indicating that they may have the ability to affect anthocyanin biosynthesis in Xiaobai’s flesh (Figure 1c–e). In addition, the expression levels of *MYB9*, *MYB10*, and *EGL3* did not show significant differences. The *MYB11* and *LWD1-like* were higher in Benihoppe, while *MYB1*, *MYB5*, *bHLH3*, *GL3*, *TTG1*, and *LWD1* were the opposite (Figure 1d), suggesting that these MBW-related TFs may also influence anthocyanin accumulation in the flesh of Xiaobai.

### 2.2. Sequence Characteristics of the FaMYB10, FaC4H, and FaTT19 in Benihoppe and Xiaobai

To explore the mutated genes in Xiaobai, we first focused on the *FaMYB10*, which was reported to play a dominant role in controlling the pigment accumulation in strawberry skin and flesh [23]. As expected, the transient overexpression of *FaMYB10* (cloned from Xiaobai) substantially restored the pigment accumulation in Xiaobai’s flesh. The overexpression of *FaMYB10* also greatly increased the anthocyanins content in the skin (Figure 1a). The qPCR analysis showed that the transcriptional level of *C4H* and *F3′H* were enhanced to more than 50-fold higher and other structural genes, including the *TT19*, *F3GT*, and *F3H,* were also increased to several-fold higher, which was consistent with the results in Benihoppe (Appendix A). However, the coding sequences of *FaMYB10* were identical in Benihoppe and Xiaobai, as confirmed by both PCR cloning and RNA-seq (Appendix A). No frameshift mutation was found in FaMYB10 (Appendix A), which was different from the white octoploid strawberry Snow Princess and the yellow diploid strawberry Yellow Wonder 5AF7 [14,24]. Since the expression level and coding sequence of *FaMYB10* did not vary in the two cultivars, and the overexpression of *FaMYB10* in Xiaobai also produced Cy3G that did not exist in Benihoppe’s flesh, this may not be the direct factor causing the absence of pigments in Xiaobai (Figure 1b and Appendix A). The coding sequences of the *FaC4H* and *FaTT19* did not produce frameshift mutations in Xiaobai, even though they contained SNP sites (Appendix A).

### 2.3. Detection of Various Flavonoid Metabolites in Strawberry Flesh by Targeted Metabolomics

To further explore the differences in the flavonoid biosynthesis pathway, we used targeted metabolomics to detect the flavonoid metabolites in the above-mentioned two flesh samples that can accumulate anthocyanins (Benihoppe and 35S::FaMYB10) and the two that cannot accumulate anthocyanins (Xiaobai and 35SN). A total of 28 flavonoid metabolites were detected, including 20 anthocyanins (Appendix A). Despite the apparent absence of pigment accumulation in Xiaobai and 35SN, there was still a certain number of flavonoid metabolites with different modifications (Table 1), indicating that they may be colorless or unstable in the flesh. Comparing Benihoppe with Xiaobai, the fold change values of 14 anthocyanin metabolites were greater than two. This was also consistent in the results of 35S::FaMYB10 and 35SN (Table 1). It was noteworthy that pelargonidin chloride and cyanidin chloride, the two initial anthocyanins in the biosynthesis pathway, were dozens of times higher in ‘Benihoppe’ (Table 1). This means that the anthocyanins in Xiaobai were repressed in the biosynthesis stage before modification, transport, and storage to vacuoles.

### 2.4. Detection of Other Metabolites in Strawberry Flesh by Quasi-Targeted Metabonomics

The above-targeted metabolomics results revealed that anthocyanins in Xiaobai’s flesh were inhibited in the flavonoid biosynthesis stage (Table 1). To search for the onset of inhibition, we used the quasi-targeted metabolomics method to further expand the detection range of metabolites. A total of 468 metabolites were detected and divided into 51 categories (Appendix A). Compared with Benihoppe, 23 differential metabolites were screened out in Xiaobai, including seven flavonoids, four phenylpropanoids and polyketides, four amino acids and their derivatives, and eight other metabolites (Appendix A). Among them, L-phenylalanine, the initial substance of flavonoid biosynthesis, was two times higher in Xiaobai than that in Benihoppe (Appendix A). Combined with the unchanged expression level of *PAL* (Figure 1c), it was suggested that repression occurs in Xiaobai’s flesh below the *PAL*. In addition, the contents of osthole, cinnamaldehyde, caffeic acid, and dihydrochalcones were also significantly increased or decreased in the phenylpropanoid biosynthesis pathway (Figure 2 and Appendix A). Although the average content of p-coumaric acid in Xiaobai was only half of that in Benihoppe, this difference was not significant due to the larger deviation. In contrast, eight-fold increases were observed in 35S::FaMYB10 (Appendix A). Combining the results of targeted metabolomics, quasi-targeted metabolomics, and qPCR, inhibited anthocyanin biosynthesis in Xiaobai was more likely to be located between PAL and ANS steps, especially in the phenylpropanoid biosynthesis pathway.

### 2.5. Whole Transcriptomic Sequencing of Red- and White-Fleshed Strawberries

To systematically identify the key genes that influence pigment accumulation in Xiaobai’s flesh, we also performed a whole transcriptome sequencing of Xiaobai, Benihoppe, 35SN, and 35S::FaMYB10. Finally, the sequencing of mRNAs, lncRNAs, and circRNAs produced 183.36 GB of the raw data. Above 93% of the reads used Q values > 30. Small RNAs sequencing generated raw data for 10.01 GB, and the Q30 values were all above 97%. In total, we obtained 109,331 mRNA transcripts (Appendix A), 17,986 lncRNA transcripts (Appendix A), 1218 circRNAs (Appendix A), 173 mature microRNA, and 185 microRNA precursors (Appendix A). To verify the reliability of the transcriptome data, we calculated the correlation between the qPCR data of the above 19 genes and their transcription levels in the transcriptome data. A strong correlation (R^2^ = 0.87) was found, indicating that the transcriptome data were highly reliable (Appendix A).

### 2.6. Analysis of the Differentially Expressed mRNAs, lncRNAs, circRNAs, and microRNAs

Among the mRNA transcripts, 881 differentially expressed genes (DEGs) were found in Benihoppe vs. Xiaobai, with 468 being up-regulated and 413 being down-regulated (Figure 3a, Appendix A). In the entire flavonoid biosynthesis pathway, the expression levels of genes including *C4H*, *CHS*, *CHI*, *F3H*, *DFR*, *ANS*, *F3GT*, and *TT19* all showed significant changes (Figure 2), which was consistent with the above qPCR results. At the same time, we also compared the DEGs in Benihoppe vs. Xiaobai and 35S::FaMYB10 vs. 35SN. Their common DEGs with the same change trend were reduced to 267, in which *C4H* showed the highest fold change in the biosynthesis pathway, indicating that the suppressed *C4H* gene may lead to the loss of anthocyanins in Xiaobai’s flesh (Figure 2). Next, the DEG functions in Benihoppe vs. Xiaobai were mainly focused on metabolic process, biological_process, single-organism metabolic process, and oxidoreductase activity by Go analysis (Figure 3b). Further KEGG analysis unraveled that the DEGs were highly enriched in the phenylpropanoid biosynthesis, but not in other pathways (Figure 3c), suggesting that the key step in repressing anthocyanin accumulation in Xiaobai’s flesh may occur in phenylpropanoid biosynthesis, especially *C4H*.

In the lncRNA transcripts, 121 were up-regulated and 104 were down-regulated in the comparison of Benihoppe vs. Xiaobai (Figure 3a, Appendix A). Only 19 differentially expressed lncRNAs showed the same change trend for Benihoppe vs. Xiaobai and 35S::FaMYB10 vs. 35SN, including *TCONS_00026735* and *TCONS_00141381*. They were upregulated by three-fold or 41-fold, respectively. They may target *CHS*, *CHI*, *ANS*, and *F3H* by co-localization and co-expression prediction methods [27]. In circRNAs, five were up-regulated and four were down-regulated in Benihoppe vs. Xiaobai (Figure 3a, Appendix A), but the log2Foldchange absolute values were all lower than two. In microRNAs, we found 80 novel matures and 96 precursors besides those already reported. Among them, seven were up-regulated and nine were down-regulated in Benihoppe vs. Xiaobai (Figure 3a, Appendix A). However, no microRNAs that could target *C4H* were found by the TargetFinder software prediction, indicating that the low expression of *C4H* in Xiaobai’s flesh may not be caused by post-transcriptional degradation.

### 2.7. Combined Analysis of Metabolomics and Transcriptomics

We further performed a combined analysis of the quasi-targeted metabolomics and transcriptomics data, and the results showed that both differential metabolites and genes in Benihoppe vs. Xiaobai were greatly enriched in the phenylpropanoid biosynthesis (Figure 3d). The expression level of *C4H* was positively correlated with the osthole, caffeic acid, and dihydrochalcones contents (Pearson correlation coefficient, r > 0.9), but negatively correlated with L-phenylalanine and cinnamaldehyde (r < −0.9), which was also consistent with their position in the pathway (Figure 2, Appendix A). In addition, *PAL* and *4CL* genes were not found to be significantly correlated with differential metabolites in Benihoppe and Xiaobai, which further reflected that the *C4H* gene was the key to the difference.

### 2.8. Overexpression of FaC4H Restored Anthocyanin Accumulation in Xiaobai’s Flesh

To verify the above hypothesis, we transiently overexpressed *FaC4H* in Xiaobai’s fruit, which was also cloned from Xiaobai. The results showed that anthocyanin accumulation was restored in the flesh (Figure 4a), and the content of Pg3G reached 206 μg/g FW (Figure 4b), which was slightly lower than Benihoppe (Figure 1b). The Cy3G was not detected, as in Benihoppe (Figure 4b). Further qPCR analysis showed that the expression levels of flavonoid-related genes, including *PAL*, *CHS*, *DFR2*, *F3GT*, *TT19*, and *F3’H,* did not increase with the accumulation of anthocyanins, while *CHI*, *F3H*, and *ANS* slightly decreased (Figure 4c). This result suggested that the repression of *FaC4H* may directly affect anthocyanin biosynthesis in Xiaobai’s flesh.

### 2.9. Methylation Detection of FaC4H and FaF3′H Promoters

The DEGs screened by transcriptome contained methyltransferase and chromatin remodeling-related genes (Appendix A). S-(5-Adenosy)-L-Homocysteine was also screened in the metabolome (Appendix A), which was the product of a methylation reaction [28]. We performed amplification and methylation detection in the promoter sequence of *FaC4H* in Benihoppe and Xiaobai. Sequence analysis showed that the *FaC4H*-promoters were basically identical, and their CpG sites were mainly located in the region from −513 bp to +345 bp with two CpG islands (Figure 5a), which were predicted online by MethPrimer. Bisulfite sequencing results revealed that all of 39 CpG sites in this region were almost unmethylated in the flesh of Benihoppe, Xiaobai, and 35S::FaMYB10 (Figure 5b). Moreover, we also explored the blocked *FaF3′H* in the flesh, which could result in undetectable Cy3G (Figure 1b). The promoter X1 sequence of *FaF3′H* shared 97% similarities with wild strawberry, while the promoter X2 sequence had a 490bp depletion from X1, accompanied by the deletion of some hormones, MYB, and circadian-related cis-acting regulatory elements (Appendix A). The methylation sequencing of the three regions with the most CpG sites on the *FaF3′H* promoters showed that a total of 50 CpG sites were almost unmethylated in Benihoppe’s skin and flesh (Appendix A), the same as those in Xiaobai and 35S::FaMYB10 (Appendix A). These results suggested that the repression of *FaC4H* and *FaF3′H* may not be caused by promoter sequence methylation.

## 3. Discussion

The biosynthesis, transport, and storage of anthocyanins in plants are jointly regulated by structural genes and related TFs in the flavonoid metabolic pathway. The loss or abnormality of their functions leads to the blocked accumulation of anthocyanins [19,21]. Xiaobai (pinkish-skinned and white-fleshed) is a bud-sport found in Benihoppe (red-skinned and red-fleshed) strawberry. Its excellent flavor and economic value make it an important candidate for cross-breeding [25]. However, the factor responsible for the white flesh of Xiaobai has not been determined to date. Both Xiaobai and Benihoppe are octoploid strawberries that are vegetatively propagated, and their genomic data are not complete and accurate enough, so it is very difficult to identify the mutation sites using QTL mapping or whole-genome sequencing [26]. Furthermore, the pigment accumulation in Xiaobai’s flesh was also not restored by directly overexpressing *FaANS*, a structural gene for anthocyanin biosynthesis [29].

In previous reports, the mutations of the *MYB10* gene were considered to be the main factor responsible for changes in anthocyanin accumulation in wild and cultivated strawberry fruits under natural conditions [14,22,23,24]. In addition to *MYB10*, mutations in the transport-related gene *PAP* (homolog of *TT19*) also lead to reduced anthocyanins in wild strawberry petioles and cultivated strawberry fruits [20]. However, we found that there was no difference in the coding sequences of *MYB10*, *RAP*, and *C4H* genes in Benihoppe and Xiaobai by RNA-seq and PCR cloning sequencing. Meanwhile, the expression levels of different *MYB10* transcripts (maker-Fvb1-3-augustus-gene-144.30, maker-Fvb1-2-snap-gene-157.15, and maker-Fvb1-1-snap-gene-139.18) also showed no significant difference between Benihoppe and Xiaobai. Although the overexpression of *FaMYB10* restored pigment accumulation in Xiaobai’s flesh, it also produced additional Cy3G, which was not present in Benihoppe’s flesh. In addition, we tried the overexpression of *FaMYB9* and *FaMYB11* in Xiaobai, which also restored anthocyanins with additional Cy3G, while the overexpression of *FaEGL3* or *FaTTG1* kept the white flesh (data not shown). These results suggested that the *FaMYB10* gene may not be directly responsible for the loss of anthocyanins in Xiaobai’s flesh. Herein, we demonstrated that the inhibition of the *C4H* gene was the main factor responsible for the absence of anthocyanins in Xiaobai, which differed from previous reports [14,23,30]. C4H not only affects the flavonoid biosynthesis but is also necessary for the biosynthesis of osthole, hydroxycinnamic acids, dihydrochalcones, eugenol, lignin, and ubiquinone [4,5,6,7]. The decline in these metabolites may be an important reason for the poor stress-resistance of Xiaobai strawberry under field cultivation conditions. The cause of the repression of *FaC4H* in Xiaobai remains a mystery.

There is evidence that the expression levels of genes are affected by transcriptional and post-transcriptional regulation, including DNA methylation, transcription factors, and microRNAs [31,32,33,34]. After excluding the repression of *FaC4H* by DNA methylation, we found a relatively highly expressed *DOF1.2* transcription factor among the screened differential genes (snap_masked-Fvb2-4-processed-gene-109.16 and augustus_masked-Fvb2-3-processed-gene-128.9), and its expression levels in Xiaobai and 35SN were about two-fold that in Benihoppe and 35S::FaMYB10 (Appendix A). The correlation calculation showed that the transcription level of *DOF1.2* was negatively correlated with the anthocyanins content (r = −0.78) and the expression of *FaC4H* (r = −0.85) in the flesh, indicating that DOF1.2 may inhibit the transcription of *FaC4H*. This negative correlation was also found in our previously reported transcriptome data [35]. Sequence analysis showed that the promoters of *FaC4H* contained 10 DOF binding sites (T/AAAAG). In Arabidopsis, AtDOF4;2 inhibits flavonoid biosynthesis by down-regulating *DFR*, *TT19*, and *LDOX* genes under cold and high light stress, but promotes the accumulation of hydroxycinnamic acids by inducing the expression of *PAL1-2*, *4CL5*, and *C4H* [36]. FcDOF4 and FcDOF16 increase the production of flavonoids in kumquat fruit by positively regulating the *C-glucosyltransferase* gene [37]. In addition, DOF TFs also play important roles in vascular cell differentiation and lignin biosynthesis, fruit ripening, biotic and abiotic stress tolerance [38]. Interestingly, the fruit-specific FaDOF2 (homolog of DOF1.2) was reported to promote eugenol synthesis together with FaEOBII by regulating *FaEGS2* expression [39,40]. After *FaDOF2* or *FaEOBII* genes are silenced in fruits, the content decreased, while anthocyanin content did not change [39,40]. However, all transcripts were barely expressed in our transcriptome data and the transcription level of *FaEGS2* was also not significantly changed, while *DOF1.2* was down-regulated after the overexpression of *FaMYB10* (Appendix A). It was unclear whether the difference between these two is caused by the strawberry varieties or the existence of a new mechanism. In addition, the other differentially expressed transcription factors such as *WRKY71* (maker-Fvb6-4-augustus-gene-286.29) and *MADS2* (augustus_masked-Fvb4-3-processed-gene-105.1), their homologous genes or family members were confirmed to be related to anthocyanin metabolism in other plant species [41,42,43]. Whether they are involved in the regulation of *FaC4H* in strawberry flesh also requires further study.

Besides the transcription factors mentioned above, the differentially expressed ncRNAs were also analyzed to determine whether they regulate *FaC4H* expression using bioinformatics methods. Unfortunately, we did not find any microRNAs or lncRNAs that could target *FaC4H*. However, the differentially expressed microRNAs fve-miR397, fve-miR399b, and fve-miR408 in Benihoppe and Xiaobai were also significantly altered in another red-fleshed cultivated strawberry Sachinoka and its white-flesh mutant [44], in which the changing trend of fve-mir399b was the same, while fve-mir397 and fve-mir408 were the opposite (Appendix A). The relationship between these microRNAs and anthocyanin metabolism in strawberry flesh still needs to be explored. Furthermore, we believed that the inhibition of *FaC4H* may not be the only factor responsible for the loss of pigment in Xiaoba’s flesh. The oxidative phosphorylation pathway, which was downstream of phenylpropanoid biosynthesis, may also have an important effect on the absence of anthocyanin accumulation in transcriptome analysis. In addition, there are many other pinkish-skinned and white-fleshed phenotypes in cultivated strawberries. It was argued that the absence of anthocyanins in their flesh is also caused by the inhibition of *C4H* expression.

Finally, the previous report showed that the main pigments in cultivated strawberries were pelargonidin compounds, while other berry species mainly accumulated cyanidin-based pigments [45]. *F3′H* was highly expressed during the whole fruit development stage in wild strawberries, with a high cyanidin content (53%), while *F3′H* sharply decreased during fruit ripening in cultivated strawberries, resulting in 88% of anthocyanins being pelargonidin [45]. Interestingly, we further found that Cy3G only existed in the skin of cultivated strawberries Benihoppe and Xiaobai, and the undetectable Cy3G in the flesh was due to the blocked *FaF3′H*. Sequence analysis showed that the coding sequence of *FaF3′H* was the same as that in wild strawberries. Although the promoter sequences of *FaF3′H* showed some differences from wild strawberries, this did not seem to be the key to blocking the expression, based on its downward trend during fruit ripening and the extremely low FPKM values of different transcripts. The methylation sequencing also revealed that the *FaF3′H* promoters were almost unmethylated in strawberry skin and flesh. Apart from that, we did not find any microRNAs that can target *FaF3′H* in the transcriptome. Therefore, we speculated that the inhibition of *FaF3′H* expression might be due to the binding of transcriptional repressors to the promoter, while ABA, high-light, or MYB10 might be able to prevent its binding and induce *FaF3′H* expression.

## 4. Materials and Methods

### 4.1. Plant Materials

The octoploid strawberry Benihoppe (HY) and its mutant cultivar Xiaobai (XB, Authorization number of plant variety rights: CNA20141360.2) were grown in a greenhouse (Sichuan Agricultural University, Chengdu, China). The environmental conditions were controlled at a temperature of 22 ± 2 °C, with artificial lighting of 220 umol∙m^−2^∙s^−1^ (cycles of 16 h light and 8 h darkness). Strawberry fruits were harvested and manually separated into the skin (outer red layer including achene) and flesh (inner layer without pith) parts, and then immediately quick-frozen with liquid nitrogen and stored at −80 °C before use.

### 4.2. Anthocyanins and PAs Detection

The main strawberry anthocyanins, pelargonidin 3-O-glucoside (Pg3G) and cyanidin 3-O-glucoside (Cy3G) were detected by the HPLC method, as we described previously [29]. Total PAs was determined using a 4-dimethylaminocinnamaldehyde assay through a full-wavelength microplate reader, as described by Prior et al. [46].

### 4.3. RT-qPCR

Total RNAs were extracted by the CTAB method [47]. The qPCR products were detected on a CFX96 real-time reaction system (Bio-Rad, Hercules, CA, USA) using the SYBR-Green (TaKaRa, Dalian, China) reagents. 26S-18S interspacer RNA was applied as a housekeeping gene and the relative quantitative data were analyzed using the 2^−ΔΔCt^ method [48]. Primers were designed in the conserved sequence regions of homoeologous copies of each gene based on the octoploid strawberry genome and transcriptome data, and are listed in Appendix A [26,29].

### 4.4. Transient Overexpression Assays in Strawberry Fruits

The coding sequences of *FaMYB10* and *FaC4H* were cloned by PCR from the Xiaobai strawberry and inserted into the expression vector pCAMBIA-35SN. Transient overexpression in strawberry fruits was conducted by following the protocols as described [49]. At least 10 fruits were selected for each replicate (three biological replicates in total).

### 4.5. Total RNA Sequencing and Analysis

Libraries of total RNA were constructed and sequenced by Novogene (Beijing, China). For lncRNAs, mRNAs, and circRNAs, the NEBNext UltraTM RNA Library Prep Kit for Illumina (NEB, Ipswich, MA, USA) was used to construct the library. For small RNAs, the NEBNext Multiplex Small RNA Library Prep Set for Illumina (NEB, USA) was used to construct the library. After quality control, all libraries were subjected to Illumina PE150 or SE50 sequencing. Three biological replicates were included for each sample. Raw reads were filtered to exclude low-quality reads and adaptors. For mRNAs, lncRNAs, and circRNAs, the clean reads were mapped and quantified using the hisat2-stringtie-bowgown pipeline [50]. The adjusted *p*-value (padj) < 0.05 and the absolute value of log2 foldchange > 1 were selected as the threshold for differentially expressed mRNAs. LncRNAs and circRNAs were identified using cuffmerge, cuffcompare, find_circ, and CIRI software, respectively, and padj < 0.05 was selected as the threshold for differentially expressed genes. The target genes of lncRNAs were predicted by the position relationship (co-location, the threshold was set to 100kb upstream and downstream of lncRNA) and expression correlation (co-expression) between lncRNAs and mRNAs [27]. For an analysis of small RNAs, reads with a length shorter than 16 nt were discarded. All reads were mapped onto the strawberry genome using bowtie [51]. The expression level of miRNAs was obtained through RPKM normalization, and padj < 0.05 was selected as the threshold for DEG detection. Mature miRNAs obtained from NGS were mapped to the sequences of miRbase v21.0 to identify known and novel miRNA in strawberries.

### 4.6. Metabolomics Detection

Libraries of metabolomics were constructed and detected by Novogene (Beijing, China) using an ExionLC™ AD system (SCIEX) coupled with a QTRAP^®^6500+ mass spectrometer (SCIEX), and the detection of experimental samples using multiple reaction monitoring was based on Novogene in-house database. In brief, 100-mg tissues were individually grounded and resuspended in 500-μL 80% methanol (volume ratio of methanol/water/formic acid was 4:1:0.001) and centrifuged at 15,000× *g*, 4 °C for 20 min. The supernatant was filtered through a 0.22-μm membrane and then diluted to a final concentration containing 53% methanol by LC-MS grade water before being injected into the HPLC-MS system. For the targeted metabolomics strategy, samples were injected into an ACQUITY UPLC HSS T3 column (100 mm × 2.1 mm) using a 7.5-min linear gradient at a flow rate of 0.2 mL∙min^−1^. The eluents were eluent A (0.1% formic acid-water) and eluent B (acetonitrile). The differentially accumulated flavonoids were screened in the same way as those with the fold change >2 and Q values < 0.05. For the quasi-targeted metabolomics method (Novogene, China), the extract was separately analyzed by the positive (BEH C8 column, 100 mm × 2.1 mm, 1.9 μm) and negative ion model (HSS T3 column, 100 mm × 2.1 mm). Samples were injected into the column at a flow rate of 0.35 mL∙min^−1^. The eluent A (0.1% formic acid–water) and eluent B (0.1% formic acid–acetonitrile) were used for positive ion model and the solvent gradient was set as follows: 5% B, 1 min; 5–100% B, 24.0 min; 100% B, 28.0 min; 100–5% B, 28.1 min; 5% B, 30 min. The eluent A (6.5mM ammonium bicarbonate–water) and eluent B (6.5mM ammonium bicarbonate-95% methanol–water) were used for negative ion model and solvent gradient was set as follows: 2% B, 1 min; 2–100% B, 18.0 min; 100% B, 22.0 min; 100–5% B, 22.1 min; 5% B, 25 min. The metabolites were annotated using the KEGG database (http://www.genome.jp/kegg/ (accessed on 10 July 2020)) and lipidmaps database (http://www.lipidmaps.org/ (accessed on 10 July 2020)). The screening threshold for different chemical accumulation in quasi-targeted metabolomics was set as the variable importance in the project (VIP) > 1.0, fold change (FC) > 1.5 or < 0.667, and the *p*-value < 0.05.

### 4.7. DNA Extraction and Bisulfite Sequencing Assays

Strawberry fruit DNA was extracted using a modified method based on Porebski et al.’s protocol [52]. Before adding the extraction buffer, the sugar-depletion buffer (1 M NaCl, 0.1 M Tris-HCl pH 8.0, 50 mM EDTA, 0.4 M glucose, 1% mercaptoethanol) was used first, at 65 °C for 10 min. Then, the mixture was centrifuged at 3000× *g* for 3 min at room temperature, and the operation was repeated once. The pellet was kept for the following steps, which were similar to Porebski et al. [52]. Promoters of *FaC4H* and *FaF3′H* were amplified from the fruit DNA in Benihoppe and Xiaobai and then ligated to a T-vector for sequencing. Fruit DNA was further processed with EpiArt^®^ DNA Methylation Bisulfite Kit (Vazyme, Nanjing, China), and primers were designed using the MethPrimer online website (http://www.urogene.org/methprimer/ (accessed on 13 September 2021)). Target sequences were amplified using a special 2× EpiArtTM HS Taq Master Mix (Vazyme, Nanjing, China), and products were ligated to the T-vector for sequencing.

### 4.8. Statistical Analysis

If not specified, all data were analyzed using the IBM SPSS Statistics 23 software, and the statistically significant differences between samples were determined using Student’s *t*-test (*p* < 0.05). Multiple comparisons were conducted using Turkey’s test and significant differences (*p* < 0.05) were indicated by different letters.

## 5. Conclusions

In this study, the pinkish-skinned and white-fleshed Xiaobai strawberry was explored to determine the molecular mechanism leading to the anthocyanin deficiency in the flesh. The transcriptional repression of the *FaC4H* in the phenylpropanoid biosynthesis pathway was responsible for the pigment loss in Xiaobai’s flesh, rather than the commonly speculated *FaMYB10*. The decrease in *FaC4H* was not directly caused by promoter methylation, lncRNAs, or microRNAs targeting, while candidate transcription factors may have a greater effect. Our findings provided a new theoretical basis for regulating the color formation of cultivated strawberry fruits.

## Figures and Tables

**Figure 1 ijms-23-07375-f001:**
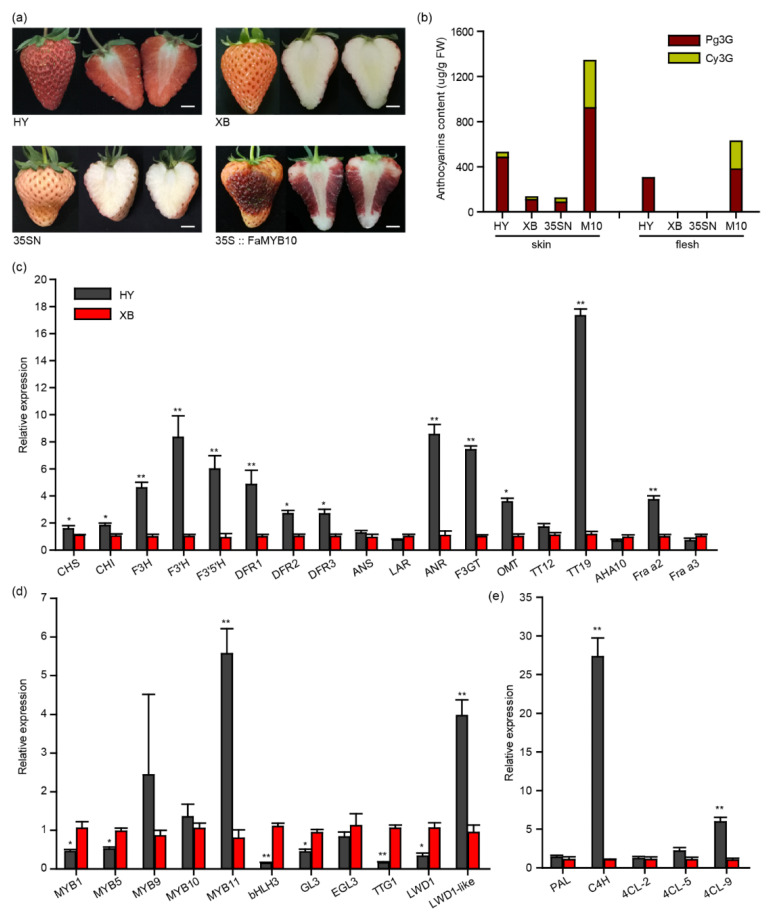
Comparison of the anthocyanins and flavonoid-related genes in the flesh of Benihoppe (HY) and Xioabai (XB) strawberries. (**a**) Ripe fruits of Benihoppe, Xioabai, and transient overexpression of *FaMYB10* in Xiaobai’s fruits (35S::FaMYB10). 35SN (empty vector) was used as a negative control. Scale bars represent 10 mm. (**b**) Pelargonidin 3-O-glucoside (Pg3G) and cyanidin 3-O-glucoside (Cy3G) were detected by HPLC in Benihoppe, Xiaobai, 35SN and 35S::FaMYB10 (M10). (**c**–**e**) Relative expression levels of structural genes or related transcription factors in flavonoid metabolism pathways. Chalcone synthase (CHS), chalcone isomerase (CHI), flavonol 3-hydroxylase (F3H), flavonol 3′-hydroxylase (F3′H), flavonoid 3′,5′-hydroxylase (F3′5′H), dihydroflavonol-4-reductase (DFR), anthocyanidin synthase (ANS), leucoanthocyanidin reductase (LAR), anthocyanidin reductase (ANR), UDP-glucose flavonoid-3-O-glycosyltransferase (F3GT), O-methyltransferase (OMT), TRANSPARENT TESTA (TT), H^+^-ATPase 10 (AHA10), Fra a allergen (Fra a), phenylalanine ammonia lyase (PAL), cinnamate 4-hydroxylase (C4H), 4-coumarate:coenzyme A ligase (4CL). Significant differences between samples were determined using Student’s *t*-test (**, *p* < 0.01; *, *p* < 0.05); multiple comparisons were conducted using Turkey’s test, and significant differences (*p* < 0.05) were indicated by different letters. Error bars show ± SEs.

**Figure 2 ijms-23-07375-f002:**
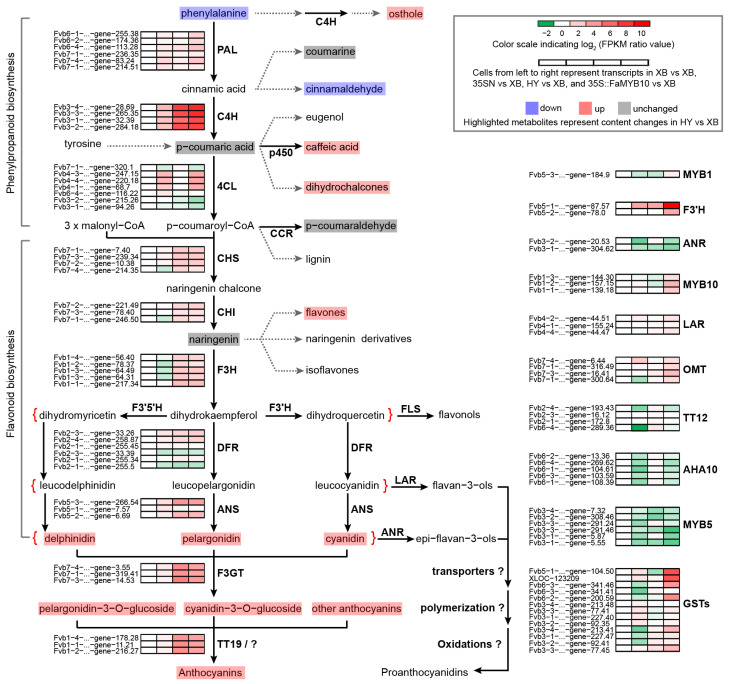
Schematic of flavonoid biosynthesis pathway in Benihoppe (HY) and Xiaobai (XB) strawberries. Four cells from the left to right represented transcripts in XB vs. XB, 35SN vs. XB, HY vs. XB, and 35S::FaMYB10 vs. XB, separately. The cell color indicated log2 FPKM ratio values from −2 (green) to 10 (red), and transcripts change trends were shown as a heatmap with abbreviated genes name on the left. The highlighted metabolites represented that they were detected by targeted metabolomics or quasi-targeted metabolomics, and the color showed the content changes in metabolites in HY vs. XB, with light blue (down), pink (up), and light grey (unchanged).

**Figure 3 ijms-23-07375-f003:**
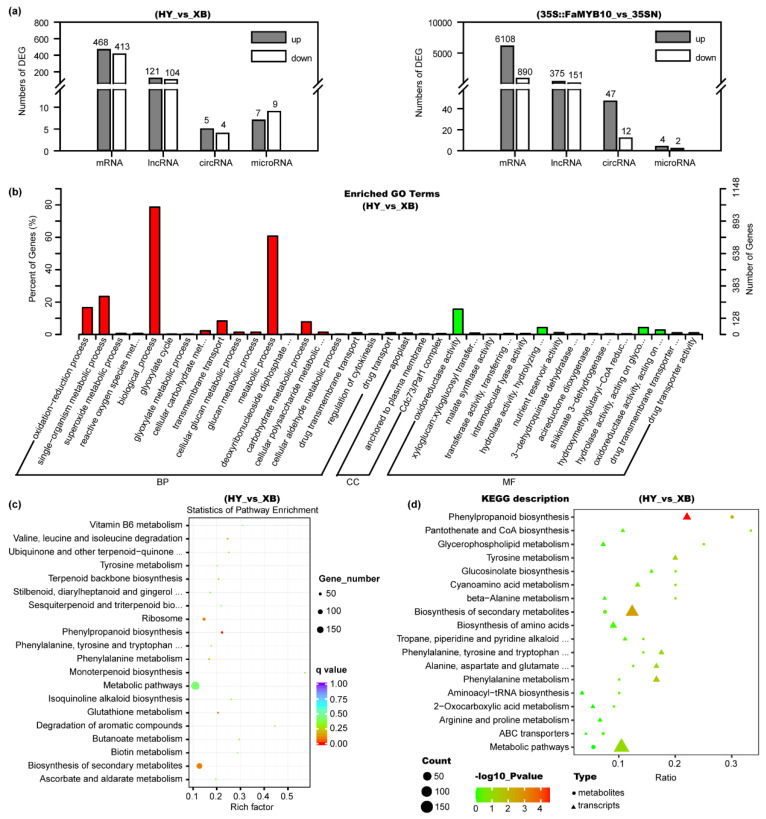
Analysis of the differential metabolites and genes in the flesh of Benihoppe (HY), Xiaobai (XB), 35S::FaMYB10, and 35SN by targeted metabolomics, quasi-targeted metabolomics, and whole transcriptomic sequencing assays. (**a**) Numbers of differentially expressed mRNAs, lncRNAs, circRNAs, and microRNAs in Benihoppe vs. Xiaobai and 35S::FaMYB10 vs. 35SN. (**b**) Function analysis of the DEGs in Benihoppe vs. Xiaobai by Gene Ontology (GO). (**c**) KEGG analysis of the DEGs in Benihoppe vs. Xiaobai. (**d**) Combined analysis of the quasi-targeted metabolomics and transcriptomics in Benihoppe vs. Xiaobai.

**Figure 4 ijms-23-07375-f004:**
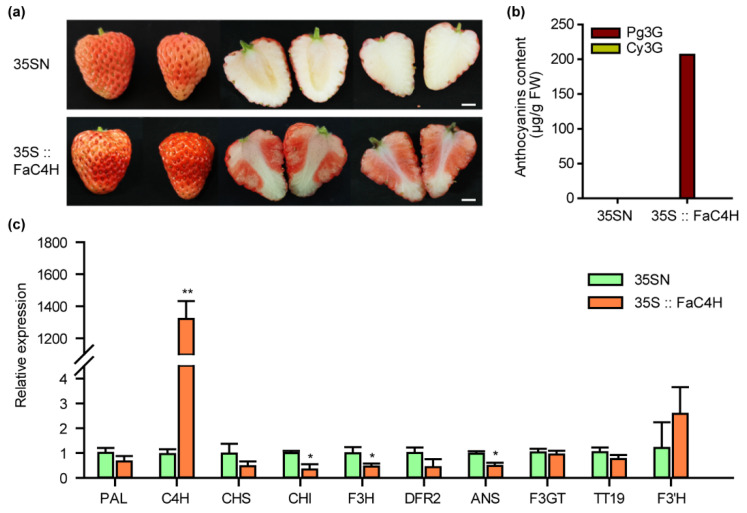
Overexpression of *FaC4H* dramatically restored anthocyanin accumulation in Xiaobai’s flesh. (**a**) Transient overexpression of *FaC4H* in Xiaobai’s fruits. 35SN was used as a negative control. Scale bars represent 10 mm. (**b**) Pg3G and Cy3G were detected in the flesh by HPLC. (**c**) Relative expression levels of structural genes in 35S::FaC4H and 35SN. Significant differences between samples were determined using Student’s *t*-test (**, *p* < 0.01; *, *p* < 0.05). Error bars show ±SEs.

**Figure 5 ijms-23-07375-f005:**
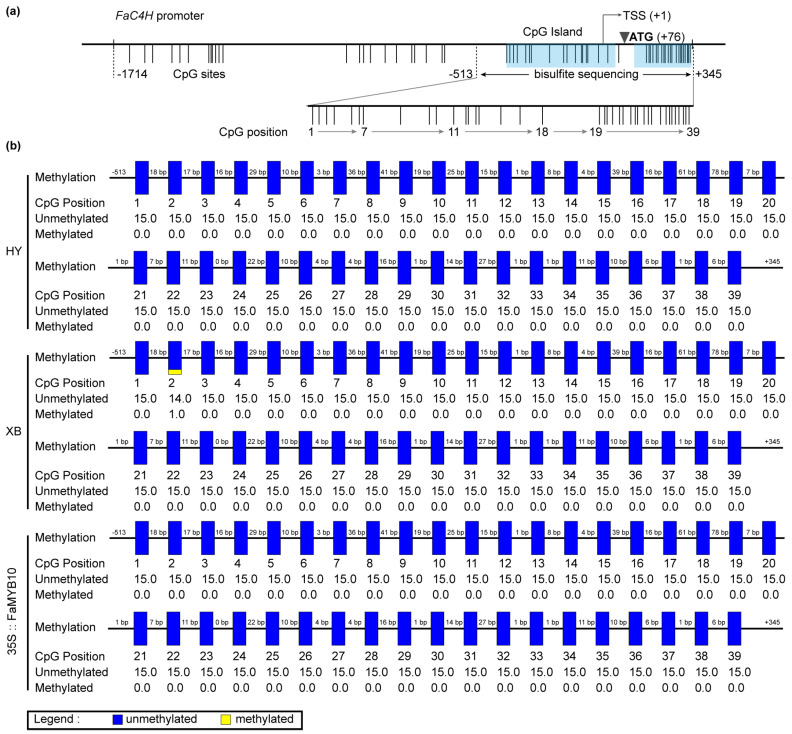
The CpG site analysis and methylation detection of *FaC4H* promoter in the flesh of Benihoppe (HY), Xiaobai (XB), and 35S::FaMYB10. (**a**) Analysis of the CpG sites in *FaC4H* promoter by MethPrimer. (**b**) Bisulfite sequencing of the selected 39 CpG sites in different samples. Each CpG site was sequenced from 15 different clones.

**Table 1 ijms-23-07375-t001:** Flavonoid compounds in the flesh of Benihoppe (HY), Xiaobai (XB), 35S::FaMYB10, and 35SN by targeted metabolomics.

No.	Compounds	Q1 ^1^ (Da)	RT ^2^ (min)	Relative Quantification ^3^				Fold Change	
				XB	35SN	HY	35S::FaMYB10	HY/XB	35S::FaMYB10/35SN
1	Pelargonidin 3-O-glucoside	449.1	1.647	(3.1 ± 1.0) × 10^4^	(3.0 ± 0.5) × 10^5^	(6.2 ± 0.4) × 10^6^	(9.9 ± 2.5) × 10^6^	199.0166	33.5122
2	Pelargonidin chloride	287.0	3.773	(1.3 ± 0.4) × 10^4^	(4.3 ± 0.4) × 10^4^	(5.0 ± 0.8) × 10^5^	(5.9 ± 0.6) × 10^5^	38.6196	13.9370
3	Pelargonidin 3,5-di-O-glucoside	611.1	1.544	0	(4.0 ± 1.4) × 10^3^	(2.8 ± 0.7) × 10^4^	(8.2 ± 1.5) × 10^4^	NA ^4^	20.3768
4	Cyanidin O-rutinoside	595.2	1.525	(1.8 ± 0.2) × 10^4^	(5.3 ± 0.4) × 10^4^	(1.4 ± 0.1) × 10^5^	(6.6 ± 0.8) × 10^6^	7.3731	123.0654
5	Cyanidin O-acetylhexoside	489.1	3.739	(5.9 ± 3.1) × 10^4^	(9.3 ± 1.2) × 10^5^	(2.9 ± 0.2) × 10^6^	(4.4 ± 1.1) × 10^6^	48.5049	4.7248
6	Cyanidin O-syringic acid	465.1	1.555	(1.2 ± 0.2) × 10^4^	(3.5 ± 0.5) × 10^4^	(4.4 ± 0.3) × 10^5^	(4.0 ± 1.0) × 10^6^	37.5528	115.2296
7	Cyanidin-3-O-galactoside chloride	465.1	1.560	(1.1 ± 0.2) × 10^4^	(3.7 ± 0.6) × 10^4^	(3.9 ± 0.2) × 10^5^	(3.9 ± 1.0) × 10^6^	34.0469	105.2074
8	Cyanidin 3-O-malonylhexoside	535.1	3.739	(2.4 ± 1.0) × 10^4^	(3.8 ± 0.3) × 10^5^	(1.0 ± 0.1) × 10^6^	(2.0 ± 0.5) × 10^6^	44.2875	5.1313
9	Cyanidin 3-O-glucoside	465.1	1.614	0	0	(4.7 ± 0.2) × 10^4^	(5.8 ± 1.6) × 10^5^	NA	NA
10	Cyanidin 3-O-rutinoside chloride	611.1	1.561	0	0	(1.5 ± 0.2) × 10^4^	(9.0 ± 1.5) × 10^4^	NA	NA
11	Cyanidin chloride	303.1	3.424	0	0	(2.4 ± 0.2) × 10^4^	(6.8 ± 1.1) × 10^4^	NA	NA
12	Delphinidin O-malonylhexoside	551.1	3.661	0	(1.8 ± 0.2) × 10^4^	(8.7 ± 0.8) × 10^4^	(1.6 ± 0.3) × 10^6^	NA	85.3311
13	Delphinidin 3-β-d-Glucoside	481.1	1.483	(2.7 ± 1.0) × 10^3^	(3.5 ± 0.2) × 10^3^	(8.8 ± 2.5) × 10^3^	(2.8 ± 0.6) × 10^4^	3.2300	7.8831
14	Peonidin chloride	317.1	3.812	0	(2.2 ± 0.4) × 10^4^	(1.1 ± 0.4) × 10^5^	(1.1 ± 0.3) × 10^5^	NA	5.1384

Q1 ^1^, molecular weight. RT ^2^, retention time. Relative quantification ^3^, calculated by the area of individual peak, and the data are expressed in mean ± standard deviation. NA ^4^, not applicable.

## Data Availability

The data that support the findings of this study are openly available in the CNGB Sequence Archive (CNSA) of China National GeneBank DataBase (CNGBdb) at https://db.cngb.org/cnsa (accessed on 13 September 2021) [53], accession number CNP0002209.

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
