# Peer review of "Alterations of Phenylpropanoid Biosynthesis Lead to the Natural Formation of Pinkish-Skinned and White-Fleshed Strawberry (Fragaria × ananassa)"

_ijms, 2022, doi:10.3390/ijms23137375_

Round 1
Reviewer 1 Report
A very interesting article, where is evidenced how the genes composition affects the anthocyanins composition. However, some minor revisions should be performed before the publication of the article.
Minor revisions:
1. The authors should add a paragraph concerning to the conclusion of this study and further studies to perform.

Author Response
A very interesting article, where is evidenced how the genes composition affects the anthocyanins composition. However, some minor revisions should be performed before the publication of the article.
Response: Thank you so much for your comments and suggestions. We carefully checked the concerns you pointed, and carefully addressed and revised them.
Minor revisions:
Point 1: The authors should add a paragraph concerning to the conclusion of this study and further studies to perform.
Response 1: Thank you for your suggestion. We added the conclusion of this study and further studies to perform in the paper. Revised portions were highlighted. Please see the attachment.
Reviewer 2 Report
Dear Editor,
I carefully read the submission titled “Alterations of phenylpropanoid biosynthesis lead to the natural formation of pinkish-skinned and white-fleshed strawberry (Fragaria ×ananassa)”. My first impression that the paper contain new information and title of the manuscript cover its content. The summary is appropriate and the aim of the work clearly established. The methods are used are adequate and used sophisticated techniques and equipment's. I found the results very reliable. Discussion and conclusions are well documented and scientifically coherent.
However I have some additions on it before acceptance.
I am of the opinion that the inclusion of L, a, b, c and h values, which are color values of fruits, in this study will contribute positively to the results of the research. These color values can be determined very easily with the colorimeter and can be interpreted with the results of this research.

Author Response
Dear Editor,
I carefully read the submission titled “Alterations of phenylpropanoid biosynthesis lead to the natural formation of pinkish-skinned and white-fleshed strawberry (Fragaria ×ananassa)”. My first impression that the paper contain new information and title of the manuscript cover its content. The summary is appropriate and the aim of the work clearly established. The methods are used are adequate and used sophisticated techniques and equipment's. I found the results very reliable. Discussion and conclusions are well documented and scientifically coherent.
Response:Thank you so much for taking your time to review this manuscript. We really appreciate all your generous comments and suggestions! According to your advice, we amended the relevant part in manuscript.
Point 1: However I have some additions on it before acceptance.
I am of the opinion that the inclusion of L, a, b, c and h values, which are color values of fruits, in this study will contribute positively to the results of the research. These color values can be determined very easily with the colorimeter and can be interpreted with the results of this research.
Response 1:Thank you for your attention and suggestion. According to your advice, we measured the L, a, b values of the skin and flesh from different strawberry samples, and presented them in Table S8. Please see the attachment.

Reviewer 3 Report
the paper is well written but it contains some grammar mistakes
the quality of the figures is very bad
the references must be updated 2019-2022
Author Response
Thank you very much for revising our manuscript. Your effort and time spent on our manuscript are greatly appreciated by all of us. Your revisions/suggestions have definitely improved the quality of our manuscript. Below are the detailed replies.
Point 1: the paper is well written but it contains some grammar mistakes.
Response 1: Thank you for your careful review. We are very sorry for the mistakes in this manuscript and inconvenience they caused in your reading. The manuscript has been thoroughly revised and rewritten by a native English speaker, so we hope it can meet the journal’s standard.
Point 2: the quality of the figures is very bad.
Response 2:Thank you for your comment and suggestion. We adjusted all the figures to make them clarity enough. Also, the sub-panels in every figure were labelled and explained. We uploaded the revised paper accompanied by all figures.
Point 3: the references must be updated 2019-2022.
Response 3:Thank you for your attention and suggestion. We carefully checked the references, and replaced some old references with new ones from 2019-2022 in the revised manuscript. We are very sorry that some references cited as experimental methods could not be successfully updated.
Round 2
Reviewer 3 Report
The authors do all corrections